# The Association of Folic Acid, Iron Nutrition during Pregnancy and Congenital Heart Disease in Northwestern China: A Matched Case-Control Study

**DOI:** 10.3390/nu14214541

**Published:** 2022-10-28

**Authors:** Ming-Xin Yan, Yan Zhao, Dou-Dou Zhao, Shao-Nong Dang, Ruo Zhang, Xin-Yu Duan, Pei-Xi Rong, Yu-Song Dang, Lei-Lei Pei, Peng-Fei Qu

**Affiliations:** 1Department of Epidemiology and Biostatistics, School of Public Health, Xi’an Jiaotong University Health Science Center, Xi’an 710061, China; 2Department of Dermatology, The First Affiliated Hospital of Xi’an Jiaotong University, Xi’an 710061, China; 3Translational Medicine Center, Northwest Women’s and Children’s Hospital, Xi’an 710061, China

**Keywords:** congenital heart disease, folic acid, iron nutrition, interaction

## Abstract

**Background**: The purpose of this study was to investigate the relationship between folic acid and iron nutrition during pregnancy and congenital heart disease (CHD) in the offspring. **Methods**: Conditional logistic regression models and nonlinear mixed-effects models were used to analyze the effects of folic acid and iron nutrition during pregnancy on CHD in offspring. **Results**: After adjusting for confounders, folic acid or iron supplementation during pregnancy reduced the risk for fetal CHD (*OR* = 0.60 (0.45, 0.82) or 0.36 (0.27, 0.48)). Similarly, dietary iron intake during pregnancy (≥29 mg/d) was associated with a reduced risk of fetal CHD (*OR* = 0.64 (0.46, 0.88)). Additionally, compared with women who only supplemented folic acid (*OR* = 0.59 (0.41, 0.84)) or iron (*OR* = 0.32 (0.16, 0.60)), women who supplemented both folic acid and iron had lower risk for newborns with CHD (*OR* = 0.22 (0.15, 0.34)). Similarly, compared with women who only supplemented folic acid (*OR* = 0.59 (0.41, 0.84)) or higher dietary iron intake (≥29 mg/d) (*OR* = 0.60 (0.33, 1.09)), women who supplemented both folic acid and higher dietary iron intake (≥29 mg/d) had lower risk for the newborn with CHD (*OR* = 0.41 (0.28, 0.62)). The combined effects were significant in the multiplication model (*OR* = 0.35 (0.26, 0.48) or 0.66 (0.50, 0.85)) but not in the additive model. **Conclusions**: Our study found that folic acid and iron nutrition during pregnancy were associated with a reduced risk of CHD in the offspring and confirmed a statistically significant multiplicative interaction between folic acid and iron nutrition on the reduced risk of CHD in offspring.

## 1. Introduction

Congenital heart disease (CHD), caused by abnormal development of the heart vessels during pregnancy, is one of the most common birth defects globally [1]. It is estimated that approximately 2–10‰ live births suffer from CHD around the world [2]. The 2015 surveillance data in China showed that CHD, with a rate of 7.6‰, has ranked first among birth defects [3]. CHD, which can lead to miscarriage, fetal death, and stillbirth, is the leading cause of neonatal deaths [4]. In China, the economic burden of new CHDs is estimated to be as high as 12.6 billion yuan per year [5]. Previous studies have reported that genetic and environmental factors may increase the risk of CHD, but a growing body of epidemiological literature suggests that a large number of non-genetic risk factors play a critical role in the development of CHD [6,7]. For example, smoking, alcohol consumption and exposure to some toxic substances during pregnancy are associated with an increased risk of CHD in offspring [8,9,10].

There is growing evidence that taking a variety of nutrients, such as folic acid and iron, during pregnancy is strongly associated with CHD [11,12]. Due to the increased nutritional requirements during pregnancy, it is acknowledged that nutrient deficiencies in this period are more likely to result in CHD [13]. An animal study found that iron deficiency during a sensitive period of embryonic development caused heart enlargement in embryonic rats [14]. A Canada cohort study found that maternal folic acid supplementation during pregnancy could significantly reduce the incidence rate of CHD in the offspring [15]. According to a cross-sectional study in China, it was found that higher dietary iron intake during pregnancy most likely reduced the risk of CHD in the offspring [16]. However, when exploring the relationship between the nutrients during pregnancy and fetal CHD, most studies have reported only one nutrient, so it is difficult to distinguish whether the effects are independent or combined.

To address this question, therefore, we conducted a matched case-control study among mothers in northwest China to achieve two objectives: (1) to explore the relationship between maternal folic acid and iron level during pregnancy and CHD in the offspring; (2) to analyze the interaction of the supplements and dietary intake of folic acid and iron during pregnancy on CHD.

## 2. Materials and Methods

### 2.1. Study Design and Participants

A matched case-control study was conducted in six birth defect surveillance hospitals in Xi’an, Shaanxi Province, Northwest China, from January 2014 to December 2016. The study was reviewed and approved by the Ethics Committee of the Xi’an Jiaotong University Health Science Center (No. 20120008), and all methods involving humans were carried out in accordance with the Declaration of Helsinki. All studied participants fully understood the study and signed an informed consent form prior to the investigation.

Cases were included according to the following inclusion criteria: gestation terminated between January 2014 and December 2016; perinatal infants (including live births and stillbirths) diagnosed with CHD according to ICD-10 classification criteria from 28 weeks of gestation to 7 days after birth at the study site, and fetuses diagnosed with CHD at less than 28 weeks of gestation in the hospital by ultrasound and other tests; fetal infants were excluded from malformations other than CHD. In the control group, singleton newborns without birth defects in the same hospital were selected according to age and date of birth in a 1:2 matching method. If a perinatal diagnosis was not yet clear or parents were unable to answer the questionnaire accurately because of psychiatric symptoms or serious illness, the subjects would be excluded.

The study sample size was determined by the rates of adequate folic acid and iron intake during pregnancy, respectively. According to 34.5% and 42% of adequate folic acid intake rates in the case and control groups, respectively, approximately 499 cases and 998 controls were required at a test level of α = 0.05 and 80% certainty [17]. A total of 338 cases and 676 controls were required based on the 31% and 40% of adequate iron intake rates in the case and control groups, respectively, at a test level of α = 0.05 and 80% certainty [18]. In our study, 603 participants were surveyed in the case group, but 3 did not complete the survey. In the control group, 1216 participants were surveyed with a no reply rate of 1.33%. In the end, 600 cases and 1200 controls with completed questionnaires were included.

### 2.2. Data Collection

A face-to-face survey using a standardized and structured questionnaire was conducted by a group of trained graduate students from the Department of Medicine, Xi’an Jiaotong University Health Science Center. The questionnaire was designed by the Xi’an Jiaotong University Health Science Center and modified based on a pilot study. The questionnaire included socio-demographic characteristics, life behavior and psychological status, dietary nutrition and nutrient supplementation during pregnancy, disease history and medication safety, environmental risk factor exposure, reproductive history and maternal health, and family genetic history. In addition to surveying the mothers themselves, their family numbers were surveyed when necessary to make the information obtained as accurate as possible. Information on the birth of newborns and diagnosis of pregnancy outcomes was collected by means of the hospital medical record. Experienced cardiovascular epidemiologists, obstetricians, pediatricians, and imaging physicians examined the questionnaire and performed clinical diagnostics of cases and controls. The specific diagnosis of CHD was made by qualified specialists based on a comprehensive analysis of the medical history, symptoms, signs and ancillary tests, of which the specialized neonatal echocardiogram and cardiovascular angiography were key points for confirming the CHD. In addition, a follow-up telephone was also undertaken within one year after birth to confirm the diagnoses.

### 2.3. Ascertainment of Study Outcomes

The outcome variable in this study was CHD, which was defined as a disorder of the heart and vascular structure and function that accompanied the birth of the children and included several different subtypes such as ventricular septal defect (VSD), atrial septal defect (ASD), atrioventricular septal defect (AVSD), patent ductus arteriosus (PDA), etc. (Appendix A) [1,19]. In this study, due to the limited sample size, only three main subtypes of CHD, including VSD, ASD, and PDA, were used for the subgroup analysis.

### 2.4. The Assessment of Nutrient Intakes during Pregnancy

Eligible women pending delivery at the hospital were interviewed via the 111-item semi-quantitative Food Frequency Questionnaire (FFQ) to collect their diet intakes throughout pregnancy. It is generally acknowledged that mothers have relatively stable dietary patterns and nutritional intake throughout pregnancy [20,21]. Therefore, the nutrient intake throughout pregnancy is representative of the nutrient intake during the 3–8 weeks of gestation (a critical period for fetal heart development). This FFQ was established based on the previous FFQ, which has been tested to have adequate validity and reliability for most nutrient intake of pregnant women in Shaanxi Province [22]. Pearson’s correlation coefficient for iron between the FFQ and the average of six 24 h recalls was 0.65, with a range of 0.53–0.70 for other nutrients. The frequency scales of food items were the eight predefined categories ranging from never to two or more times per day, and women were asked to recall and identify their portion sizes according to food portion images. Daily intakes of dietary iron and folic acid were transformed using the China Food Composition Tables [23].

Dietary iron was calculated and divided into two groups (<29 mg/d, ≥29 mg/d) based on the recommended level of iron intake during the whole pregnancy [24]. The recommended intake of folic acid during pregnancy is 600 μg per day, and folic acid supplementation is about 400 μg per day. Dietary folic acid is calculated based on the recommended intake of folic acid during pregnancy and divided into two groups (<200 μg/d, ≥200 μg/d) [25]. Pregnant women were asked to recall the type/brand, duration days, and amount of nutritional supplements they used during the first trimester through the early trimester. We defined taking folic acid for more than 90 days before and during the first trimester as taking folic acid supplements and taking iron before and during the first trimester for more than 30 days as taking iron.

### 2.5. Assessment of Socio-Demographic and Health-Related Characteristics

The socio-demographic and health-related characteristics in this study were selected based on previous CHD-related studies [8,9,26,27]. The study information mainly included: (1) Socio-demographic characteristics: maternal age (<30 years and ≥30 years), maternal ethnicity (Han and Other), maternal education (Senior high school or lower and College or above), residence (rural and urban), Wealth index (Poor, Moderate and Rich), (2) health-related characteristics: colds in early pregnancy (Yes, No), fever in early pregnancy (Yes, No), alcohol consumption in pregnancy (Yes, No), passive smoking in pregnancy (Yes, No), hair dyeing in pregnancy (Yes, No) and pregnancy frequency (1, ≥2).

Alcohol consumption included a variety of alcoholic beverages (e.g., white wine, beer, red wine, etc.) during the whole pregnancy. Passive smoking was defined as inhaling smoke for more than 15 min per day and at least one day per week during pregnancy; Hair dyeing and perming was defined as dyeing or perming hair ≥ 1 time during pregnancy. We used principal component analysis to construct a wealth index for measuring household economic level, incorporating variables such as monthly household income, monthly expenditure, housing type, household appliances, and transportation [28]. This wealth index was divided into thirds as an indicator for the poor, medium, and rich households.

### 2.6. Statistical Analysis

In the first step, the study population characteristics and energy-adjusted nutrient intakes by the residual method were described according to case and control groups. The categorical variables were expressed as frequencies (n) and percentages (%) and were compared between groups using the χ^2^ test. Secondly, three conditional logistic regression models were developed to explore the relationship between folic acid, iron nutrition, and CHD with several levels of adjustment as follows: Model 1 was a crude model without any covariates; Model 2 adjusted for age, ethnicity, education, residence, wealth index, cold in early pregnancy, fever in early pregnancy, drinking, passive smoke, hair dyeing and perming, gravidity based on Model 1; Model 3 adjusted for folic acid supplementation, iron supplementation, dietary iron intake, dietary folic acid intake based on Model 2.

Thirdly, the multiplicative interaction of folic acid and iron nutrition on the incidence of CHD was explored with the multiplicative action term in Model 2. Meanwhile, we also developed the additive model to explore the additive interaction using a nonlinear mixed effects model [29]. The synergy index (S), attributable proportion (AP), and relative excess risk due to interaction (RERI) were used to assess biological interactions in additive models. S is the ratio of the combined effect to the sum of the individual effects. AP shows the proportion of the interaction effect in the total effect. RERI refers to the difference between the sum of the joint effect and the individual effect. When the confidence interval of RERI and AP does not contain 0, and the confidence interval of S does not contain 1, then there is an additive interaction.

A series of sensitivity analyses were performed to check the robustness of our findings. First, we assessed the interaction of folic acid and iron nutrition on the incidence of three different subtypes of CHD, including VSD, ASD, and PDA, using the same methodology. Second, we performed the same analysis by the subgroups of covariates (age, education, residence, and wealth index) to explore the stability of the model. All statistical analyses were completed using SAS version 9.4 (SAS Institute, Cary, NC, USA), *p* < 0.05 indicated a significant difference.

## 3. Results

### 3.1. Participants’ Characteristics

A total of 1800 subjects were included in this study, including 600 cases in the case group and 1200 cases in the control group. The statistically significant differences in baseline characteristics of mothers between the case and control group were found (Table 1), including ethnicity, education, residence, wealth index, cold in early pregnancy, drinking, passive smoke, hair dyeing and perming, and gravidity. There were no statistically significant differences in maternal age and fever in early pregnancy between the two groups.

### 3.2. The Association of Folic Acid or Iron Nutrition with CHD

Table 2 shows the relationship between folic acid and iron nutrition during pregnancy and CHD in the offspring. The crude analysis (Model 1) suggested that folic acid and iron supplementation and dietary iron and folic acid intake were associated with a lower risk for CHD among infants. Both Models 2 and 3 can be regarded as finally adjusted, depending on whether or not folic acid supplementation, iron supplementation, dietary iron intake, and dietary folic acid intake are viewed as part of the confounding structure. Results were similar in Model 2 and 3 and suggested a significant negative relationship except for dietary folic acid intake; folic acid and iron supplementation and higher dietary iron intake were associated with lower CHD risk. After adjusting for all baseline covariates, folic acid supplementation, iron supplementation, dietary iron intake, and dietary folic acid intake, Model 3 showed that the risk of CHD was negatively correlated with folic acid supplementation (*OR* = 0.60 (0.45, 0.82)), iron supplementation (*OR* = 0.36 (0.27, 0.48)), and dietary iron intake (≥29 mg/d) (*OR* = 0.64 (0.46, 0.88)) during pregnancy, respectively. However, we did not find any association between dietary folic acid intake during pregnancy and fetal CHD.

### 3.3. Interaction Effect Analysis

After adjusting for potential confounders, Table 3 shows that pregnant women who are supplemented with both folic acid and iron significantly have a reduced risk of giving birth to a baby with CHD (*OR* = 0.22 (0.15, 0.34)). Compared to those who did not supplement with folic acid and iron, the risk of CHD was lower in women who only supplemented with iron (*OR* = 0.32 (0.16, 0.60) or folic acid (*OR* = 0.59 (0.42, 0.83)). Similarly, mothers who had folic acid supplementation and higher dietary iron intake during pregnancy together were less likely to have a neonate with CHD compared to those who had lower dietary iron intake and did not supplement folic acid (*OR* = 0.41 (0.28, 0.62)). Table 4 presents the multiplicative and additive interaction of folic acid and iron nutrition on CHD. After adjusting for potential confounders, the multiplicative interaction of folic acid supplementation with iron supplementation in pregnancy concerning the reduced risk of CHD in the offspring (*OR* = 0.35 (0.26, 0.48)) was significant. Similarly, the multiplicative interaction of folic acid supplementation and dietary iron intake in the incidence of CHD in the offspring (*OR* = 0.66 (0.50, 0.85)) was also found. To investigate the additive interaction of folic acid and iron nutrition, the previously mentioned indicators RERI, AP, and S were used. The additive interaction values were as follows: S = 0.71 (0.55, 0.92), AP = 1.44 (−0.23, 3.11), RERI = 0.32 (0.02, 0.61) for folic acid supplementation and iron supplementation, and S = 0.73 (0.45, 1.17), AP = 0.54 (−0.56, 1.63), and RERI = 0.22 (−0.19, 0.63)) for folic acid supplementation and dietary iron intake. However, it was observed that almost all the indicators for folic acid supplementation and iron nutrition did not reach a statistically significant level.

### 3.4. Sensitivity Analysis

When different CHD subgroups were used as the outcome variables, the multiplicative interaction of folic acid supplementation and iron nutrition was still significant, while the additive interaction index remained statistically insignificant (Appendix A). Stratified by different ages, educations, residences and wealth indices, the risk of fetal CHD was significantly lower among mothers who were supplemented with both folic acid and iron compared to those who were not supplemented (Appendix A). According to the multiplicative interaction of folic acid supplementation and iron supplementation, the results were statistically significant by different demographic characteristics subgroups (all *p* < 0.05). According to S, AP, and RERI, all subgroup analysis of additive interaction was mostly essentially unchanged, without statistically significant difference (*p* > 0.05) (Appendix A). Furthermore, when stratified by different demographic characteristics, the interaction of supplemental folic acid and dietary iron intake was not materially different (Appendix A).

## 4. Discussion

Our study investigated the association of maternal folic acid and iron nutrition during pregnancy with the risk of CHD in the offspring and explored the interaction of maternal folic acid and iron nutrition in incidence of CHD, using a matched case-control design in Northwest China. We have got two key findings. One, mothers who supplemented with folic acid or iron during pregnancy reduced the risk of giving birth to newborns with CHD. Similarly, more than 29 mg/d dietary intake of iron during pregnancy reduced the risk of CHD in their offspring. Two, we also found the multiplicative interaction of folic acid supplementation with iron supplementation in pregnancy concerning the reduced risk of CHD in the offspring. Meanwhile, the multiplicative interaction of folic acid supplementation and dietary iron intake in the incidence of CHD in the offspring was also observed. However, the additive interaction of folic acid supplementation and iron nutrition still needed to be confirmed. The sensitivity analysis of the study also showed stable results.

First, maternal demand for nutrients gradually increases during pregnancy, but a large number of women do not get enough nutrients during this period [13]. Due to fetal development, in particular, women during pregnancy need 3–6 times higher folic acid than those in the non-pregnant state. Xu et al. found that maternal folic acid supplementation during pregnancy reduced the risk for CHD by 40% based on the meta-analysis [30]. In a Dutch case-control study, mothers who supplemented folic acid during pregnancy had a reduced risk of fetuses with CHD [31]. A similar result from the birth cohort was also obtained in China [32]. Studies in Canada have shown that during pregnancy, food fortification with folic acid could lead to a 6% reduction in the type of severe CHD per year [33]. Folic acid is an important carrier source in the metabolism of one-carbon units in the body. When the body is deficient in folic acid, the body conserves one-carbon units through regulatory mechanisms, which may lead to hyperhomocysteinemia [34]. A high level of homocysteine is considered to be an independent risk factor for CHD, which in turn affects embryonic heart development [34,35]. Our findings support the importance of folic acid supplementation during pregnancy for the prevention of CHD in women of childbearing age. Nevertheless, the association of dietary folic acid intake with CHD was not confirmed yet.

Second, iron’s nutritional status also plays a key role in the development of the cardiovascular system. In China, however, the prevalence of iron deficiency among women of childbearing age is 49.6%, reaching 61.7% among pregnant women [36]. Previous research found that iron deficiency can cause tissue hypoxia and stress response, which can affect the normal development of the cardiovascular system [37]. It was reported that maternal iron supplementation or higher dietary iron intake during pregnancy reduced the risk of CHD in the fetus [38]. In animal experiments, iron deficiency was associated with heart enlargement during the sensitive period of cardiovascular development in embryonic rats [14]. A case-control study consisting of 410 CHD cases and 100 controls in China found differences in iron levels between these two groups [39]. Our findings support the importance of both iron supplementation and higher dietary iron intake during pregnancy for the prevention of CHD in offspring.

Third, we determined the combined effect of maternal concomitant folic acid and iron nutrition during pregnancy on the incidence of CHD in the offspring. After adjusting for potential confounders, the multiplicative interaction of folic acid with iron nutrition during pregnancy concerning the reduced risk of CHD in the offspring was significant. In other words, folic acid administration and higher dietary iron intake during pregnancy synergistically contributed to the lower incidence of CHD in the offspring. Previous studies have found that folic acid supplementation and phenytoin therapy have a superimposed effect on better seizure control [40]. Folic acid supplementation and RFC1 (GG) genotype during pregnancy have gene-nutrition interactions on fetal neural tube defects (NTDs) [41]. The mechanism of action of folic acid and iron nutrition interacting with offspring CHD may be complex. According to an animal study, the low iron status may alter the utilization of folic acid despite adequate folic acid intake throughout the organism [42]. However, further studies will be required to corroborate these findings and, if confirmed, to elucidate possible reasons for these changes.

Our study is the first to explore the interaction between folic acid and iron nutrition during pregnancy in the incidence of CHD in the offspring, providing valuable evidence for the prevention of CHD in the future. However, some limitations of our study are worth discussing. First, the information on dietary and non-dietary during pregnancy was obtained through the mother’s own recall, which might lead to recall bias. Second, the dietary information throughout pregnancy, rather than during the critical period of fetal cardiac development (3–8 weeks of gestation) was recalled by mothers using FFQ. However, studies have suggested that mothers have stable dietary patterns and nutritional intake throughout pregnancy [18,19]. Third, due to the small sample size, we cannot determine the additive interaction of folic acid and iron nutrition when interpreting our results. Finally, these findings are based on case-control studies, and the causal relationship between maternal folic acid and iron nutrition during pregnancy and CHD in the offspring could not be demonstrated. Another large-sample population-based cohort study should be established subsequently to confirm these relationships.

In conclusion, this study found that folic acid supplementation, iron supplements and higher dietary iron intake during pregnancy reduced the risk of CHD. In particular, the multiplicative interaction of folic acid supplementation and iron nutrition in the incidence of CHD was confirmed, and these two nutrients synergistically contributed to the lower incidence of CHD in the offspring. Therefore, from a maternal and child health perspective, sufficient folic acid and iron nutrients during pregnancy would help to reduce fetal CHD. 

## Figures and Tables

**Table 1 nutrients-14-04541-t001:** Characteristics of participants in the case group and the control group.

Variables	Cases (N = 600)	Controls (N = 1200)	*χ* ^2^	*p* Value
Age, n (%)			3.085	0.079
<30 years	378 (63.00)	806 (67.17)		
≥30 years	222 (37.00)	394 (32.83)		
Ethnicity, n (%)			10.658	0.001
Han	585 (97.50)	1192 (99.33)		
Other	15 (2.50)	8 (0.67)		
Education, n (%)			88.615	<0.001
Senior high school or lower	264 (44.00)	270 (22.50)		
College or above	336 (56.00)	930 (77.50)		
Residence, n (%)			178.249	<0.001
Urban	201 (33.50)	800 (66.67)		
Rural	399 (66.50)	400 (33.33)		
Wealth index, n (%)			116.335	<0.001
Poor	313 (52.17)	331 (27.58)		
Moderate	167 (27.83)	398 (33.17)		
Rich	120 (20.00)	471 (39.25)		
Cold in early pregnancy, n (%)			22.227	<0.001
No	415 (69.17)	951 (79.25)		
Yes	185 (30.83)	249 (20.75)		
Fever in early pregnancy, n (%)			3.161	0.075
No	546 (91.00)	1120 (93.33)		
Yes	54 (9.00)	80 (6.67)		
Drinking, n (%)			15.298	<0.001
No	582 (97.00)	1192 (99.33)		
Yes	18 (3.00)	8 (0.67)		
Passive smoke, n (%)			36.090	<0.001
No	255 (42.50)	690 (57.50)		
Yes	345 (57.50)	510 (42.50)		
Hair dyeing and perming, n (%)			13.292	<0.001
No	563 (93.83)	1168 (97.33)		
Yes	37 (6.17)	32 (2.67)		
Gravidity, n (%)			24.396	<0.001
1	236 (39.33)	620 (51.67)		
≥2	364 (60.67)	580 (48.33)		

**Table 2 nutrients-14-04541-t002:** The association of maternal folic acid and iron nutrition during pregnancy with CHD.

Variables	Cases n (%)	Controls n (%)	Model 1 ^a^ OR (95% CI)	Model 2 ^b^ OR (95% CI)	Model 3 ^c^ OR (95% CI)
Folic acid supplementation					
No	141 (23.50)	164 (17.67)	1.00	1.00	1.00
Yes	459 (76.50)	1036 (86.33)	0.52 (0.40, 0.67)	0.62 (0.47, 0.84)	0.60 (0.45, 0.82)
Iron supplementation					
No	514 (85.67)	756 (63.00)	1.00	1.00	1.00
Yes	86 (14.33)	444 (37.00)	0.29 (0.23, 0.38)	0.37 (0.28, 0.49)	0.36 (0.27, 0.48)
Dietary iron intake					
<29 mg/d	434 (72.33)	721 (60.08)	1.00	1.00	1.00
≥29 mg/d	166 (27.67)	479 (39.92)	0.57 (0.45, 0.70)	0.69 (0.54, 0.89)	0.64 (0.46, 0.88)
Dietary folic acid intake					
<200 μg/d	421 (70.17)	771 (64.25)	1.00	1.00	1.00
≥200 μg/d	179 (29.83)	429 (35.75)	0.76 (0.62, 0.94)	0.82 (0.64, 1.05)	1.09 (0.79, 1.50)

^a^ No factors were adjusted. ^b^ Model 2 used Model 1 and adjusted for age, ethnicity, education, residence, wealth index, cold in early pregnancy, fever in early pregnancy, drinking, passive smoke, hair dyeing and perming, and gravidity. ^c^ Adjusted for Model 2 and/or folic acid supplementation, iron supplementation, dietary iron intake, and dietary folic acid intake.

**Table 3 nutrients-14-04541-t003:** Interaction effect of maternal folic acid and iron nutrition on the occurrence of CHD.

Variables	Folic Acid Supplementation ^c^	N n (%)	Cases n (%)	Controls n (%)	Model 1 ^d^ OR (95% CI)	Model 2 ^e^ OR (95% CI)
Iron supplementation ^a^						
0	0	228 (12.67)	120 (20.00)	108 (9.00)	1.00	1.00
1	0	77 (4.28)	21 (3.50)	56 (4.67)	0.34 (0.19, 0.60)	0.32 (0.16, 0.60)
0	1	1042 (57.89)	394 (65.67)	648 (54.00)	0.54 (0.41, 0.73)	0.59 (0.42, 0.83)
1	1	453 (25.16)	65 (10.83)	388 (32.33)	0.15 (0.11, 0.22)	0.22 (0.15, 0.34)
Dietary iron intake ^b^						
0	0	209 (11.61)	109 (18.17)	100 (8.33)	1.00	1.00
1	0	96 (5.33)	32 (5.33)	64 (5.33)	0.42 (0.25, 0.70)	0.60 (0.33, 1.09)
0	1	946 (52.56)	325 (54.17)	621 (51.75)	0.47 (0.34, 0.64)	0.59 (0.41, 0.84)
1	1	549 (30.50)	134 (22.33)	415 (34.58)	0.28 (0.20, 0.40)	0.41 (0.28, 0.62)

^a^ 0 = No and 1 = Yes. ^b^ 0 = “<29 mg/d”. 1 = ”≥29 mg/d”. ^c^ 0 = No and 1 = Yes. ^d^ No factors were adjusted. ^e^ Model 2 used Model 1 and adjusted for age, ethnicity, education, residence, wealth index, cold in early pregnancy, fever in early pregnancy, drinking, passive smoke, hair dyeing and perming, gravidity.

**Table 4 nutrients-14-04541-t004:** Multiplicative and additive interaction of maternal folic acid and iron nutrition in incidence of CHD.

Variables	Interactions	Model 1 ^d^	Model 2 ^e^
Multiplication model	Folic acid supplementation × Iron supplementation	0.26 (0.20, 0.35)	0.35 (0.26, 0.48)
Additive model	RERI ^a^	0.27 (0.02, 0.53)	0.32 (0.02, 0.61)
	S ^b^	0.76 (0.61, 0.94)	0.71 (0.55, 0.92)
	AP ^c^	1.79 (−0.25, 3.83)	1.44 (−0.23, 3.11)
Multiplication model	Folic acid supplementation × Dietary iron intake	0.54 (0.43, 0.68)	0.66 (0.50, 0.85)
Additive model	RERI	0.40 (0.14, 0.66)	0.22 (−0.19, 0.63)
	S	0.64 (0.52, 0.80)	0.73 (0.45, 1.17)
	AP	1.42 (0.20, 2.64)	0.54 (−0.56, 1.63)

^a^ The relative excess risk due to interaction. ^b^ The attributable proportion. ^c^ The synergy index. ^d^ No factors were adjusted. ^e^ Model 2 used Model 1 and adjusted for age, ethnicity, education, residence, wealth index, cold in early pregnancy, fever in early pregnancy, drinking, passive smoke, hair dyeing and perming, gravidity.

## Data Availability

The data present in this study are available on request from the corresponding authors.

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
