# Peer review of "The Association of Folic Acid, Iron Nutrition during Pregnancy and Congenital Heart Disease in Northwestern China: A Matched Case-Control Study"

_nutrients, 2022, doi:10.3390/nu14214541_

Round 1

Reviewer 1 Report

Comments:

1. Could the authors please provide the ethical approval for their study please?

2. Could the authors please clarify how the numbers of experiments were determined i.e. how the study was powered please?

3. Could the authors please clarify the statistical analyses please?

Reviewer 2 Report

In this study Authors evaluated the impact of several nutrition variables including folic acid assumption in 600 pregnancies complicated by a fetus with CHD and compared to a control group with a match ratio of 1:2

The subject is of interest and I would like to congratulate with Authors for their effort.

My questions are

1) how many questionaries Authors delivered to reach the 600 (study group) and 1200 (control group) size. In other words the % of no reply should be given

2)the type of CHD should be listed in a separate table

3)details on how the CHS were identified is necessary

Round 2

Reviewer 2 Report

Authors revised their manuscript as requested